# Classification and Antigen Molecules of Autoimmune Bullous Diseases

**DOI:** 10.3390/biom13040703

**Published:** 2023-04-20

**Authors:** Takashi Hashimoto, Hua Qian, Norito Ishii, Takekuni Nakama, Chiharu Tateishi, Daisuke Tsuruta, Xiaoguang Li

**Affiliations:** 1Department of Dermatology, Osaka Metropolitan University Graduate School of Medicine, Osaka 545-8585, Japan; 2Department of Laboratory Medicine, Medical College, Dalian University, Dalian 116622, China; 3Department of Dermatology, Kurume University School of Medicine, Kurume 830-0011, Japan

**Keywords:** autoimmune bullous diseases, autoantibody, classification, autoantigen molecules, subtypes, pemphigus, pemphigoid, nomenclature

## Abstract

Autoimmune bullous diseases (AIBDs), which are a group of tissue-specific autoimmune diseases of the skin, present with various blistering lesions on the skin and mucous membranes, and show autoantibodies of IgG, IgA and IgM against epidermal cell surfaces and basement membrane zone. To date, AIBDs have been classified into a number of distinct subtypes by clinical and histopathological findings, and immunological characteristics. In addition, various biochemical and molecular biological studies have identified various novel autoantigens in AIBDs, which has resulted in proposals of new subtypes of AIBDs. In this article, we summarized various distinct AIBDs, and proposed the latest and most comprehensive classification of AIBDs with their autoantigen molecules.

## 1. Introduction

Autoimmune bullous diseases (AIBDs) can be divided into two major groups, pemphigus diseases with anti-epidermal cell surface antibodies (intraepidermal AIBDs) and pemphigoid diseases with anti-epidermal basement membrane zone (BMZ) autoantibodies (subepidermal AIBDs). AIBDs are rare tissue-specific autoimmune diseases of the skin and the autoantibodies have been proved to be pathogenic and develop the mucocutaneous lesions in AIBDs.

Most AIBDs are refractory and severe, and are treated with a number of different therapies, including steroids, immunosuppressive drugs, plasmapheresis, high-dose intravenous immunoglobulin and biological agents. As the responses to treatments vary among the disease subtypes, a reliable diagnosis is very important for the selection of appropriate treatment. The diagnosis is made by the results of immunological and biochemical tests, such as immunofluorescence, immunoblotting and ELISA, in addition to clinical and histological findings.

To date, various biochemical and molecular biological studies have identified a number of distinct autoantigens in AIBDs, and accordingly, novel diagnostic methods have also been developed. Based on these results, new subtypes of AIBDs have been proposed, and a new classification of AIBDs has also been proposed [1,2]. The general classification of AIBDs is described in previously published papers [1] and textbooks. In this article, we would like to summarize all subtypes of AIBDs, and propose the latest and most comprehensive classification of AIBDs, as well as the corresponding autoantigen molecules.

## 2. Classification of Pemphigus Diseases (Intraepidermal AIBDs)

The pemphigus diseases clinically present with blistering and erosive lesions on the skin and mucous membranes, with histopathological evidence of intraepidermal acantholytic blister formation. The blistering lesions are developed within the epidermis, which is located in the outermost layer of the skin, and autoantibodies against the cell surface proteins of epidermal cells are proved to directly induce the lesions. Various desmosomal constituent proteins present on the cell surfaces of keratinocytes, the major cell component of the epidermis, are the major autoantigens for pemphigus diseases. Desmoglein 3 (Dsg3) and Dsg1 are most frequently detected as autoantigens of pemphigus diseases, although some other autoantigens have recently been identified [1].

The pemphigus diseases are usually divided into classical and non-classical groups of pemphigus (Table 1). There are four major subtypes in the classical pemphigus group, while the non-classical pemphigus group includes a variety of relatively rare forms, such as pemphigus herpetiformis, drug-induced pemphigus, paraneoplastic pemphigus (PNP), atypical PNP, anti-plakin dermatosis, anti-desmocollin (Dsc) dermatosis, intercellular IgG/IgA dermatosis (IgG/IgA pemphigus), intercellular IgA dermatosis (IAD, IgA pemphigus) and IgM pemphigus (Table 1).

### 2.1. Classical Pemphigus Disease Group (Table 1)

Classical pemphigus can be divided into four main types, pemphigus vulgaris (PV) and its subtype pemphigus vegetans, and pemphigus foliaceus (PF) and its subtype pemphigus erythematosus.

PV has three subtypes, including mucosal dominant-type, mucocutaneous-type and cutaneous-type. Mucosal dominant-type PV presents erosive lesions on the oral mucosa, particularly on the buccal mucosa, tongue and lips; whereas mucocutaneous-type PV presents, in addition to erosive lesions on the oral mucosa, flaccid blisters and refractory erosions on the skin of entire body, particularly on the intertrigo areas, such as axilla and groin. Cutaneous-type PV presents only skin lesions, without lesions on mucous membranes.

Pemphigus vegetans presents vegetating skin eruptions, such as verrucous and nodular lesions. Pemphigus vegetans has two subtypes, Neumann type, which begins with a PV-like blistering and erosive skin rash, and Hallopeau type, which begins with a pustular skin rash. PV and pemphigus vegetans are in general more severe than PF and pemphigus erythematosus.

PF presents smaller and superficial blisters, mainly on the trunk. Pemphigus erythematosus presents with a relatively mild PF-like skin rash with facial erythema, which resembles butterfly-rash seen in systemic lupus erythematosus (SLE). Patients of pemphigus erythematosus often show positive antinuclear antibodies. Oral mucosal lesions are absent in PF and pemphigus erythematosus.

Mucosal dominant-type PV reacts only with Dsg3, whereas mucocutaneous-type and cutaneous-type PV react with both Dsg3 and Dsg1. Pemphigus vegetans reacts with Dsg3 and Dsg1, and may also react with Dsc1 and Dsc3 [3]. PF and pemphigus erythematosus react with only Dsg1. However, there have been reports of oral mucosal-type PF, which reacts only with Dsg1 but shows oral mucosal lesions [4,5].

### 2.2. Non-Classical Pemphigus Disease Group (Table 1)

In addition to the classical pemphigus disease group described above, various non-classical pemphigus diseases have been proposed. In these non-classical pemphigus diseases, a number of autoantigens other than Dsg, particularly Dsc1-3, have also been identified [3,6].

Pemphigus herpetiformis clinically presents with annular and herpetiform erythematous skin lesions with blister formation, mimicking dermatitis herpetiformis. Pemphigus herpetiformis reacts mainly with Dsg1, but also occasionally with Dsg3 and Dsc1-3 [3].

Drug-induced pemphigus is induced by various drugs, most frequently by d-penicillamine, and its autoantibodies recognize mainly with Dsg1, and less with Dsg3 or other autoantigens [7,8].

PNP is mainly associated with hematological malignancies, most frequently malignant lymphomas, and presents with Stevens-Johnson syndrome-like severe mucosal lesions, mainly oral and ocular mucosae, as well as a variety of skin lesions. PNP has a poor prognosis and may be fatal, particularly when complicated by bronchiolitis obliterans [9]. The major autoantigens of PNP are all currently known plakin family proteins (plectin, epiplakin [10], desmoplakin I/II, BP230, envoplakin and periplakin), with anti-envoplakin and anti-periplakin antibodies being of high diagnostic value [11]. PNP reacts with many other autoantigens, including Dsg3, Dsg1, Dsc1-Dsc3, BP180 and alpha-2-macroglobulin-like-1 (A2ML1), and recently we have found that autoantibodies against transglutaminase 1 (TG1) are detected specifically in PNP and have high diagnostic value [12]. Atypical PNP patients may present no oral mucosal lesions or detect no neoplasms [11,13].

Recently, Oiso et al. reported a case clinically showing lichen planus-like skin rash, positive for both anti-envoplakin and anti-periplakin antibodies, but not associated with malignancy, and proposed a novel disease entity, anti-plakin dermatosis [14].

In addition, IgG and IgA antibodies against Dsc1-3 have been detected in various pemphigus disease groups, and only anti-Dsc antibodies have been detected in some cases [3]. Although the disease entity has not yet been established and the pathogenicity of anti-Dsc antibodies is currently not completely elucidated, we propose the new disease entity, anti-Dsc pemphigus, for these cases [3]. A monoclonal antibody against the extracellular domain of Dsc3 could cause intraepidermal blistering in a model of human skin, and a loss of intercellular adhesion in cultured keratinocytes [15]. Dsc3-reactive IgG antibodies purified from sera of pemphigus patients could induce the loss of adhesion of epidermal keratinocytes in a dispase-based keratinocyte dissociation assay [16]. These findings supported that IgG autoantibodies against Dsc3 may contribute to blister formation in pemphigus. By using an active disease mouse model (adoptive transfer of Dsc3 lymphocytes to Rag2−/− immunodeficient mice that express Dsc3 and Dsg3), the presence of anti-Dsc3 autoantibodies is sufficient to induce the appearance of a pathological phenotype relatable to atypical pemphigus [17]. In addition, anti-Dsc3 antibodies in sera of pemphigus patients were proved to mainly recognize the extracellular 2 domain of Dsc3, and antibodies against the extracellular 2 domain of Dsc3 have pathogenic roles in keratinocyte dissociation [18]. It is also suggested that the induction of pathogenic anti-Dsc3 IgG is associated with Dsc3-specific T cells that recognize Dsc3 in association with HLA-DRB1∗04:02 [19]. Although these studies proved the possible involvement of anti-Dsc3 antibodies in the pathogenesis of pemphigus, more deep studies are necessary to elucidate it clearly.

Cases presenting simultaneously with IgG and IgA anti-epidermal cell surface antibodies have been reported as IgG/IgA pemphigus. We have proposed the name of intercellular IgG/IgA dermatosis for these cases, in line with the proposed name of IAD, as described below [20]. Autoantigens specific to these cases have not currently been identified, but autoantibodies against Dsc3 are often detected [20].

We proposed the name IAD as a more appropriate name for the group of diseases that have been referred to as IgA pemphigus [21,22]. We also proposed a tentative classification of IAD with five subtypes, including subcorneal pustular dermatosis (SPD)-type, intraepidermal neutrophilic IgA dermatosis (IEN)-type, PV-type, PF-type and unclassified-type (Table 1) [21,22].

Recently, Emtenani et al. generated IgA anti-Dsg3 monoclonal antibodies by the phage display method using B cells from patients with PV-type IAD. Their pathogenic study using experimental models such as skin organ culture showed that the IgA monoclonal autoantibodies could induce FcR-dependent neutrophil migration, leading to the development of skin lesions [23]. This is different from the pathogenesis in conventional IgG pemphigus, which shows acantholysis without the involvement of neutrophils.

In 2017, Keiny et al. reported an unusual smoldering Waldenstrom’s macroglobulinaemia patient, which was suggestive of IgM pemphigus based on the IgM deposition around keratinocytes in direct and indirect immunofluorescence assays [24]. Negative reactivity with Dsg1, Dsg3, envoplakin or periplakin in either IB or ELISA studies indicated that the IgM antibodies in this patient might react with so far unidentified antigen(s) [24].

## 3. Classification of Pemphigoid Disease (Subepidermal AIBD) Groups (Table 2)

The pemphigoid diseases with anti-epidermal BMZ autoantibodies show mucocutaneous blistering lesions with a histopathological finding of subepidermal blisters. There are a number of distinct pemphigoid diseases which react with a variety of autoantigens, the constituent proteins of the hemidesmosomes and related structures in the epidermal BMZ (see Table 2).

Bullous pemphigoid (BP) reacts mainly with two autoantigens (BP180 and BP230), while IgG antibodies against the NC16a domain of BP180 are considered to be most pathogenic. BP classically presents with itchy edematous erythema and large tense blisters. However, some BP cases, particularly in the early stages, may show an erythematous, wheal-like, eczematous skin lesions without blisters, although direct or indirect immunofluorescence assays show positive results. The terms of non-bullous BP [25] and prodromal BP [26] have been proposed for these BP cases. Some research groups object to making a diagnosis of ‘bullous’ pemphigoid in cases that do not present with blisters, and the consensus on the diagnostic name for these cases will require further expert discussion.

BP is further classified into a number of subtypes based on its characteristic clinical features, including localized BP, nodular BP, BP vegetans, eczema-like BP, vesicular pemphigoid, dyshidrosiform pemphigoid, bullous lichen planus and lichen planus pemphigoides (Table 2).

Lichen planus pemphigoides is characterized clinically by lichen planus lesions with numerous bullae, and immunologically by linear deposits of IgG and/or C3 at BMZ in peri-lesional skin and antoantibodies reactive with the NC16a and C-terminal domains of BP180 and BP230 [27,28].

Drug-induced pemphigoid is also known, and has recently attracted particular attention in relation to dipeptidyl peptidase-4 (DPP-4) inhibitors [29] and immune checkpoint inhibitors [30,31]. The autoantibodies in drug-induced pemphigoid react mainly with BP180 (NC16a domain, LAD-1 and C-terminal domain) and less with BP230 [29,30,31]. Radiation-induced BP is also known, and appeared mostly after the cessation of radiation treatment, and some of the cases were identified to be positive for autoantibodies against BP180 and BP230 [32,33]. Some burn-induced BP have also been reported, with antibodies against the non-NC16a domain of BP180 [34].

There have also been reports of BP cases showing only anti-BP230 antibodies without anti-BP180 antibodies, and we proposed the name “anti-BP230-type BP” for this condition [35]. Although autoantibodies against BP230, the intracellular protein, have been considered non-pathogenic, a recent study using skin specific-BP230-deficient mice showed the pathogenic potential of anti-BP230 antibodies [36]. Therefore, the pathogenesis of anti-BP230-type BP needs to be further elucidated in the future.

In addition to IgG anti-epidermal BMZ autoantibodies, many cases with IgE anti-epidermal BMZ autoantibodies have been reported, and the autoantigen and pathogenicity of IgE autoantibodies have been investigated [37,38,39]. Although the disease name “IgE BP” has not been used, it is tentatively listed in Table 2 to describe this disease entity.

Pemphigoid gestationis (herpes gestationis) is considered to be BP arising during pregnancy or the postpartum period. IgG autoantibodies against the NC16a domain of BP180 could be found in patient sera, and their titers are well correlated with disease activities along the disease course [40].

Mucous membrane pemphigoid (MMP) (previously called cicatricial pemphigoid; however, as many cases do not leave scars, the name “MMP” is now encouraged to be used) is a subtype of AIBDs with bullous and erosive lesions, mainly on oral, ocular and other mucous membranes. The most common type is anti-BP180-type MMP with IgG and IgA antibodies against the C-terminal domain of BP180 [41], followed by anti-laminin 332-type MMP showing IgG anti-laminin 332 antibodies [42].

Pure ocular MMP showing exclusively the ocular mucosal lesions has been reported to show IgG and IgA anti-integrin β4 autoantibodies [43], which were also detected by immunoblotting using the hemidesmosome-enriched fraction [44]. Oral MMP showing exclusively oral mucosal lesions has been reported to show IgG anti-integrin α6 autoantibodies [45].

Recently, our group reported a unique MMP case with oral mucosal lesions, which was positive for IgA and IgG autoantibodies against p200 (laminin γ1), but negative for all known MMP-correlated autoantibodies [46]. Therefore, p200 (laminin γ1) is a new autoantigen of MMP, and this case was considered as the first report of anti-p200 (laminin γ1)-type MMP [46]. This study and literature review further supported the potential contribution of anti-p200 (laminin γ1) autoantibodies on mucosal lesions [46,47]. Antibodies against laminin α5 were also first reported in an MMP case, although its roles in MMP is still undetermined [46]. Two independent cases of MMP with IgM anti-BMZ antibodies were reported under the diagnosis of IgM MMP [48,49]. In 2023, Philip et al. reported 93 cases of ocular MMP with IgM anti-BMZ antibodies [50], although the autoantigens were not disclosed. Therefore, IgM MMP deserves more attention.

Cases with a typical clinical presentation of MMP, which showed IgA-dominant anti-BMZ antibodies, have occasionally been reported under the diagnosis of linear IgA disease (LAD) (details are described below). However, we believe that only the cases clinically showing typical skin lesions should be diagnosed as LAD, and the cases with predominant mucosal involvement should be diagnosed as IgA-dominant MMP.

Epidermolysis bullosa acquisita (EBA) presents with refractory cutaneous and oral mucosal lesions similar to the dystrophic type of epidermolysis bullosa hereditaria and shows IgG anti-type VII collagen autoantibodies [51].

Bullous SLE develops bullous skin lesions with high SLE activity. Bullous SLE is classified into two conditions; i.e., bullous SLE in narrow sense, which shows IgG anti-type VII collagen autoantibodies, and bullous SLE in broad sense, which does not show autoantibodies and develops blisters due to the liquefaction degeneration of epidermal BMZ [52].

In 1996, we originally described anti-p200 pemphigoid as a novel disease entity reacting with an unknown 200 kDa antigen in two reports [53,54]. We subsequently suggested laminin γ1 as an autoantigen, and suggested the name “anti-laminin γ1 pemphigoid” [55]. Anti-p200 pemphigoid is usually responsive to treatments. Japanese patients with anti-p200 pemphigoid have been reported to be frequently associated with psoriasis [56], although this is not the case in western countries. In addition, atypical anti-p200 pemphigoid with nail involvement and blisters over the joints was also reported [57].

Dermatitis herpetiformis (Duhring) is common in Europe and the USA, particularly in Scandinavia, although this disease is rarely reported in Japan. This difference is thought to be due to genetic differences such as HLA. Dermatitis herpetiformis mainly presents with eczema-like vesicular skin lesions on the elbows, knees and buttocks, with a histopathological finding of microabscesses in the papillary dermis, and direct immunofluorescence shows the granular deposition of IgA just below the epidermis [58]. Circulating IgA antibodies to TG3 (epidermal TG) and TG2 are detected [58]. Dermatitis herpetiformis is considered as a cutaneous lesion of coeliac disease [58].

LAD presents clinically with annular erythema and vesiculobullous lesions, and histologically with microabscesses in the papillary dermis, similar to dermatitis herpetiformis. However, LAD shows linear IgA deposition in the epidermal BMZ by direct immunofluorescence. The patient sera are positive for IgA anti-epidermal BMZ autoantibodies. From the results of IgA indirect immunofluorescence using 1M NaCl-split skin, LAD is classified into lamina lucida (LL)-type LAD, which reacts with the epidermal side of the split skin, and sublamina densa (SD)-type LAD, which reacts with the dermal side of the split skin.

The disease has also been reported under the name linear IgA bullous dermatosis (LABD). However, similar to the discussion of BP above, because there are cases of LAD that show predominantly erythematous lesions without blisters, we advocate using LAD without ‘bullous’ rather than LABD [59].

Some groups use the diagnosis of IgA EBA when IgA anti-type VII collagen antibodies are detected in SD-type LAD patients. However, considering the characteristic clinical presentation of EBA with IgG anti-type VII collagen antibodies and the historical background of the nomenclature of LAD and EBA, we proposed that such cases should be diagnosed as SD-type LAD rather than IgA EBA [59]. Table 3 shows a new LAD classification, with corresponding autoantigens, which we currently proposed [59].

The name “linear IgA/IgG bullous dermatosis (LAGBD)” has been proposed for the disease that simultaneously shows IgG and IgA anti-epidermal BMZ antibodies [60]. However, the disease entity LAGBD is not yet universally accepted, as IgG anti-BMZ antibodies are known to coexist in LAD, and IgA anti-epidermal BMZ antibodies may be shown in BP. Further investigation of the autoantigens and pathogenesis of LAGBD is awaited.

Cases showing IgM anti-epidermal BMZ antibodies have occasionally been reported [61], although their diagnosis and autoantigen were not clarified. Cases with IgM autoantibodies with type VII collagen have previously been reported [62,63]. Recently, Bock et al. [64] and Hirano et al. [65] reported cases with IgM anti-BMZ autoantibodies, which were suggested to react with BP180, and proposed to diagnose the cases as IgM pemphigoid. Future analysis of this disease awaits.

We analyzed 20 cases with clinical and histopathological findings resembling dermatitis herpetiformis, which showed the granular deposition of C3 and C5b-9 in the BMZ by direct immunofluorescence. These cases were negative for other complement components and immunoglobulin deposition, and no circulating autoantibodies were detected. We considered these cases as a new disease entity, and proposed the name “granular C3 dermatosis (GCD)” [66]. However, the pathogenesis of the disease and the mechanism of C3 deposition in the skin remain unclear. Recently, we reported a case of possibly drug-induced GCD [67]. However, because the GCD-like granular deposition of C3 may also be observed in other diseases [68], further clinical investigations and basic research of GCD are needed to clarify the disease entity and the mechanism of granular C3 deposition.

## 4. Ending

In the present article, to show a comprehensive classification of AIBDs, multiple subtypes of AIBDs were summarized according not only to the differences of known AIBDs-correlated autoantibodies, but also the differences of clinical features and inducements. All known autoantigens and immunoglobulin types of corresponding autoantibodies for each subtype of AIBDs were also collected in this article. However, questions might be raised on such a detailed and complex classification of AIBDs. Is it necessary to, for example, divide BP into drug-induced BP, radiation-induced BP, burn-induced BP and so on? Does it have any consequences, for example, a difference in therapy, or does it have academic value? Some of the detailed classification of AIBDs in the present article might not be useful for physicians’ clinical practices currently; however, the comprehensive classification may benefit analyses of potential disease inducement, the identification of unknown AIBDs-correlated autoantibodies, studies on potential pathogenesis mechanisms, investigating why some treatments work or do not work in some patients with AIBDs, and possible re-classifications with novel findings in the future; all of which will finally enhance our understanding on AIBDs, particularly the effects of autoantibodies, and improve clinical practices on the diagnosis of and therapy for AIBDs.

To show various types of AIBDs comprehensively, we also presented our ideas on the nomenclature of some types of AIBDs; for example, we prefer to name it SD-type LAD but not IgA EBA for patients positive for IgA anti-type VII collagen antibodies, although a consensus on the final diagnostic names for some cases of AIBDs still needs further expert discussion.

Recently, some studies reported the occurrence of AIBDs after the administration of coronavirus disease 2019 (COVID-19) vaccination [69,70,71,72]; however, currently there are not yet enough data to confirm the correlation between COVID-19 vaccination and increased risk of AIBDs [73].

There is a wide variety of AIBDs with autoantibodies with a number of different cutaneous autoantigens, each of which has a different pathogenesis and effective treatment. Therefore, a reliable diagnosis of AIBDs by the detection of autoantibodies to various autoantigens is important for the early selection of appropriate treatment for individual patients.

Currently, ELISA kits for the detection of autoantibodies against Dsg1, Dsg3, BP180, BP230 and type VII collagen are commercially available. However, it has recently been reported that the chemiluminescent enzyme immunoassay (CLEIA) methods for Dsg1, Dsg3 and BP180, which are currently used in clinical practice, sometimes show false negative results due to the short reaction time with patient sera in CLEIA system [74]. In such cases, various other serological tests, including ELISA, should be used in combination with CLEIAs.

In the future, it is desirable to develop a comprehensive diagnostic method that may enable the diagnosis of all AIBDs by the identification of all autoantigens.

With the disease progression, one subtype of AIBDs might shift to another subtype, or concur with other subtype(s) of AIBDs [75,76]. Therefore, during therapy, it is necessary to repeatedly test autoantibodies and autoantigens, which may suggest a change of treatment regimens. In addition, the significant involvement of IgA autoantibodies during the disease progression of some AIBDs, such as MMP, should also be emphasized [77].

This article presented an extensive classification of almost all currently known AIBDs and their subtypes. The disease concepts for some diseases have not yet been established. We hope that future studies of autoantigens and pathogenesis will clarify the independence of these diseases, leading to the establishment of the final classification of AIBDs and their subtypes.

## Figures and Tables

**Table 1 biomolecules-13-00703-t001:** Classification, immunoglobulin types and autoantigens of pemphigus diseases.

Classification		Immuno- globulin Types	Autoantigens
Classical pemphigus groups			
Pemphigus vulgaris (PV)			
	Mucosal dominant-type	IgG	Desmoglein 3 (Dsg3)
	Mucocutaneous-type	IgG	Dsg3, Dsg1
	Cutaneous-type	IgG	Dsg3, Dsg1
Pemphigus Vegetans			
	Neumann-type	IgG	Dsg3, Dsg1, desmocollin 1 (Dsc1), Dsc3
	Hallopeau-type	IgG	Dsg3, Dsg1, Dsc1, Dsc3
Pemphigus foliaceus (PF)		IgG	Dsg1
	Oral mucosal-type	IgG	Dsg1
Pemphigus erythematosus		IgG	Dsg1
Non-classical pemphigus groups			
Pemphigus herpetiformis		IgG	Dsg1, Dsg3, Dsc1-3
Drug-induced pemphigus		IgG	Dsg1
Paraneoplastic pemphigus (PNP)		IgG	Plectin, epiplakin desmoplakin I/II, BP230, envoplakin, periplakin, BP180, alpha-2-macroglobulin-like-1(A2ML1), Dsg3, Dsg1, Dsc1-3, transglutaminase 1 (TG1)
Atypical PNP		IgG	Same to PNP’s
Anti-plakin dermatosis		IgG	Envoplakin, periplakin
Anti-Dsc pemphigus		IgG/IgA	Dsc1-3
Intercellular IgG/IgA dermatosis (IgG/IgA pemphigus)		IgG/IgA	Undetermined or Dsc1-3 or multiple detections
Intercellular IgA dermatosis (IgA pemphigus)			
	Subcorneal pustular dermatosis (SPD)-type	IgA	Dsc1
	Intraepidermal neutrophilic IgA dermatosis (IEN)-type	IgA	Undetermined
	PV-type	IgA	Dsg3
	PF-type	IgA	Dsg1
	Unclassified-type	IgA	Undetermined or multiple detected
IgM pemphigus		IgM	Undetermined (?)

**Table 2 biomolecules-13-00703-t002:** Classification, immunoglobulin types and autoantigens of pemphigoid diseases.

Classification		Immunoglobulin Types or Complements	Autoantigens
Bullous pemphigoid (BP) and its subtypes			
	BP	IgG	BP230, BP180 (NC16a domain)
	Non-bullous BP/Prodromal BP (skin rash, erythema, wheals/urticaria)	IgG	BP230, BP180 (NC16a domain)
	Localized BP	IgG	BP180
	Nodular BP	IgG	BP180, etc
	BP vegetans	IgG	BP180, etc
	Eczema-like BP	IgG	BP180, etc
	Vesicular pemphigoid	IgG	BP180, etc
	Dyshidrosiform pemphigoid	IgG	BP180, etc
	Bullous lichen planus	IgG	No autoantibodies
	Lichen planus pemphigoides	IgG	BP180 (NC16a and C-terminal domains), BP230
	Drug induced-BP		
	DPP-4 inhibitor-associated	IgG	BP180 (central domain)
	Immune checkpoint inhibitor-associated	IgG	BP180
	Radiation-induced BP	IgG	BP180, BP230
	Burn-induced BP	IgG	BP180, BP230
	Anti-BP230-type BP	IgG	BP230
	IgE BP	IgE	BP180, BP230
	Pemphigoid (herpes) gestationis	IgG	BP180 (NC16a domain)
Mucous membrane pemphigoid (MMP) and its subtypes			
	Anti-BP180-type MMP	IgG/IgA	BP180 (C-terminal domain)
	Anti-laminin 332-type MMP	IgG	Laminin 332
	Pure ocular MMP	IgG/IgA	Integrin β4
	Anti-p200 (laminin γ1)-type MMP	IgA/IgG	p200 (laminin γ1)
	Oral MMP	IgG	Integrin α6
	IgM MMP	IgM	Undetermined
Epidermolysis bullosa acquisita		IgG	Type VII collagen
Bullous systemic lupus erythematosus (SLE)			
	Bullous SLE in narrow sense	IgG	Type VII collagen
	Bullous SLE in broad sense	IgG	No autoantibodies
Anti-p200 pemphigoid		IgG	p200 (laminin γ1)
Atypical anti-p200 pemphigoid		IgG	p200
Dermatitis herpetiformis (Duhring)		IgA	Transglutaminase (TG)3, TG2
Linear IgA disease (LAD)			
	Lamina lucida (LL)-type LAD	IgA	97kDa/120kDa LAD-1, etc.
	Sublamina densa (SD)-type LAD	IgA	Type VII collagen, etc.
Linear IgA/IgG Bullous dermatosis (LAGBD)		IgA/IgG	Undetermined or multiple detections
IgM pemphigoid		IgM	BP180 (?), type VII collagen (?)
Granular C3 dermatosis		C3	No autoantigen (?)

**Table 3 biomolecules-13-00703-t003:** Classification and autoantigens of linear IgA disease (LAD).

Classification	Immunoglobulin Types	Autoantigens
Anti-LAD97 LL-type LAD	IgA	97 kDa LAD1
Anti-LAD1 LL-type LAD	IgA	120 kDa LAD1
Anti-BP180 NC16a domain LL-type LAD	IgA	BP180 NC16a domain
Anti-Type VII collagen SD-type LAD	IgA	Type VII collagen
Anti-p200 SD-type LAD	IgA	p200 (laminin γ1)

LL, lamina lucida; SD, sublamina densa.

## Data Availability

The original contributions presented in this study are included in the article. Further inquiries can be directed to the corresponding authors.

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
