# Peer review of "Classification and Antigen Molecules of Autoimmune Bullous Diseases"

_biomolecules, 2023, doi:10.3390/biom13040703_

Round 1
Reviewer 1 Report
In the present manuscript "classification and antigen molecules of autoimmune bullous diseases" authors provide comprehensive review of multiple phenotypes of AIBD and involved antigens. I find manuscript well written, however, I would like to ask authors to discuss more on the necessity of this complex classification of AIBD. Is it not an overcomplication? Is it necessary to, for example, divide bullous pemphigoid into drug-induced BP, radiation-induced BP and burn-induced BP? Does it have any consequences (for example, difference in therapy), or it is more of academic question? As I have the feeling, that complex classification without clear implementation of the differences will not be understood by physicians and will not be used.
English language is fine, just some sentences could be improved and rephrased.
Author Response
Comments of reviewer 1
(1) In the present manuscript "classification and antigen molecules of autoimmune bullous diseases" authors provide comprehensive review of multiple phenotypes of AIBD and involved antigens. I find manuscript well written, however, I would like to ask authors to discuss more on the necessity of this complex classification of AIBD. Is it not an overcomplication? Is it necessary to, for example, divide bullous pemphigoid into drug-induced BP, radiation-induced BP and burn-induced BP? Does it have any consequences (for example, difference in therapy), or it is more of academic question? As I have the feeling, that complex classification without clear implementation of the differences will not be understood by physicians and will not be used.
Reply: We agree to this comment. However, the purpose of this manuscript is to show all types of AIBDs comprehensively. Therefore, we hope that the reviewer will allow us to show these contents and tables. For the clinical purposes for the practitioners, we would like to prepare more concise one in the next paper.
According to the suggestion of reviewer, we also added a discussion, which is also shown below.
“In the present article, to show a comprehensive classification of AIBDs, multiple subtypes of AIBDs were summarized, according to not only the differences of known AIBDs-correlated autoantibodies, but also the differences of clinical features and inducements. All known autoantigens and immunoglobulin types of corresponding autoantibodies for each subtype of AIBDs were also collected in this article. However, questions might be raised on such a detailed and complex classification of AIBDs. Is it necessary to, for example, divide BP into drug-induced BP, radiation-induced BP, burn-induced BP and so on? Does it have any consequences, for example, difference in therapy, or does it have an academic value? Some of the detailed classification of AIBDs in the present article might not be useful for physicians’ clinical practices currently, however, the comprehensive classification may benefit for analyses of potential disease inducement, identification of unknown AIBDs-correlated autoantibodies, studies on potential pathogenesis mechanisms, investigating why some treatments work or not in some patients with AIBDs, and possible re-classifications with novel findings in the future, all of which will finally enhance our understanding on AIBDs, particularly the effects of autoantibodies, and improve clinical practices on diagnosis and therapy of AIBDs.
To show various types of AIBDs comprehensively, we also presented our ideas on nomenclature of some types of AIBDs, for example, we prefer to name it SD-type LAD, but not IgA EBA for patients positive for IgA anti-type VII collagen antibodies, although the consensus on the final diagnostic names for some cases of AIBDs still needs further expert discussion. ”
(2) English language is fine, just some sentences could be improved and rephrased.
Reply: As the reviewer suggested, we have checked through the text, and improved the English language.
Reviewer 2 Report
1. What is the main question addressed by the research?
The classification of autoimmune bullous diseases.
2. Do you consider the topic original or relevant in the field? Does it
address a specific gap in the field?
The authors present their suggestion on updating the classification of autoimmune bullous diseases.
3. What does it add to the subject area compared with other published
material?
The comprehensive new classification system
4. What specific improvements should the authors consider regarding the
methodology? What further controls should be considered?
No need to improve methodology, no controls are required as this is not an experimental manuscript
5. Are the conclusions consistent with the evidence and arguments presented
and do they address the main question posed?
Conclusions are fine
This manuscript deals with the issue of inadequacy of the current nomenclature of autoimmune bullous diseases affecting the skin and mucous membranes. This issue is gaining recognition from researchers in the field. So, the ideas of the authors are not novel. Therefore, I think it is absolutely necessary to include the recent paper (Front Immunol. 2022 Dec 19;13:1103375. doi: 10.3389/fimmu.2022.1103375. eCollection 2022) on this topic in the reference list as ref. 2 in the introduction section at the end of this sentence: Based on these results, new subtypes of AIBDs have been proposed, and a new classification of AIBDs has also been proposed [1].
Author Response
Comments of reviewer 2
This manuscript deals with the issue of inadequacy of the current nomenclature of autoimmune bullous diseases affecting the skin and mucous membranes. This issue is gaining recognition from researchers in the field. So, the ideas of the authors are not novel. Therefore, I think it is absolutely necessary to include the recent paper (Front Immunol. 2022 Dec 19;13:1103375. doi: 10.3389/fimmu.2022.1103375. eCollection 2022) on this topic in the reference list as ref. 2 in the introduction section at the end of this sentence: Based on these results, new subtypes of AIBDs have been proposed, and a new classification of AIBDs has also been proposed [1].
Reply: As the reviewer suggested, this paper has been cited as reference 2 in the revised manuscript.
Reviewer 3 Report
The manuscript "Classification and antigen molecules of autoimmune bullous diseases" submitted by Takashi Hashimoto et al aims to review the literature on autoimmune bullous diseases and their classification based on autoantigens involved and immunoglobulin isotypes.
It is well written and interesting to read.
I agree with the authors that it would be necessary to develop an “universal” diagnostic method and tools that may enable to diagnose all AIBD, worldwide. Too often, we realize that Elisa, or IF, or Western Blot, or cell based assay… have been used to establish the diagnosis, not to mention the origin of the recombinant proteins used: E coli, baculovirus, mammalian cells and whether the proteins are conformational or not. This does not help comparison of case report and published studies.
I think this manuscript is worth publishing and I thank the authors for their work on this review.
I have however a major comments:
I was a little surprised by one of the statement p4, line 118:
“… and the pathogenicity of anti-Dsc antibodies is currently unknown…”
I know of literature on anti DSC3 that has shown to be pathogenic in vitro by the authors, including by some of this manuscript authors (see below). Could you specify what you meant in this sentence?
J Invest Dermatol . 2021 Sep;141(9):2123-2131.e2. doi: 10.1016/j.jid.2021.01.032. Epub 2021 Mar 22.
Autoantibodies to DSC3 in Pemphigus Exclusively Recognize Calcium-Dependent Epitope in Extracellular Domain 2
Hiroshi Koga 1, Kwesi Teye 2, Yoshihiko Otsuji 3, Norito Ishii 3, Takashi Hashimoto 4, Takekuni Nakama 3
Am J Pathol. 2011 Feb;178(2):718-23. doi: 10.1016/j.ajpath.2010.10.016.
IgG autoantibodies against desmocollin 3 in pemphigus sera induce loss of keratinocyte adhesion
David Rafei 1, Ralf Müller, Norito Ishii, Maria Llamazares, Takashi Hashimoto, Michael Hertl, Rüdiger Eming
Author Response
Comments of reviewer 3
The manuscript "Classification and antigen molecules of autoimmune bullous diseases" submitted by Takashi Hashimoto et al aims to review the literature on autoimmune bullous diseases and their classification based on autoantigens involved and immunoglobulin isotypes.
It is well written and interesting to read.
I agree with the authors that it would be necessary to develop an “universal” diagnostic method and tools that may enable to diagnose all AIBD, worldwide. Too often, we realize that Elisa, or IF, or Western Blot, or cell based assay… have been used to establish the diagnosis, not to mention the origin of the recombinant proteins used: E coli, baculovirus, mammalian cells and whether the proteins are conformational or not. This does not help comparison of case report and published studies.
I think this manuscript is worth publishing and I thank the authors for their work on this review.
(1) I have however a major comment:
I was a little surprised by one of the statement p4, line 118:
“… and the pathogenicity of anti-Dsc antibodies is currently unknown…”
I know of literature on anti DSC3 that has shown to be pathogenic in vitro by the authors, including by some of this manuscript authors (see below). Could you specify what you meant in this sentence?
J Invest Dermatol . 2021 Sep;141(9):2123-2131.e2. doi: 10.1016/j.jid.2021.01.032. Epub 2021 Mar 22.
Autoantibodies to DSC3 in Pemphigus Exclusively Recognize Calcium-Dependent Epitope in Extracellular Domain 2
Hiroshi Koga 1, Kwesi Teye 2, Yoshihiko Otsuji 3, Norito Ishii 3, Takashi Hashimoto 4, Takekuni Nakama 3
Am J Pathol. 2011 Feb;178(2):718-23. doi: 10.1016/j.ajpath.2010.10.016.
IgG autoantibodies against desmocollin 3 in pemphigus sera induce loss of keratinocyte adhesion
David Rafei 1, Ralf Müller, Norito Ishii, Maria Llamazares, Takashi Hashimoto, Michael Hertl, Rüdiger Eming
Reply: We thank the reviewer for this important comment and kindly providing these 2 references. We have revised the above description and also added some new sentences to describe the details on the potential pathogenesis of anti-Dsc antibodies by citing total 5 new references including the 2 references provided by the reviewer. The revised and newly added sentences were also shown below.
“Although the disease entity has not yet been established and the pathogenicity of anti-Dsc antibodies is currently not completely elucidated, we propose the new disease entity, anti-Dsc pemphigus, for these cases [3]. A monoclonal antibody against the extracellular domain of Dsc3 could cause intraepidermal blistering in a model of human skin, and a loss of intercellular adhesion in cultured keratinocytes [15]. Dsc3-reactive IgG antibodies purified from sera of pemphigus patients could induce loss of adhesion of epidermal keratinocytes in a dispase-based keratinocyte dissociation assay [16]. These findings supported that IgG autoantibodies against Dsc3 may contribute to blister formation in pemphigus. By using an active disease mouse model (adoptive transfer of Dsc3 lymphocytes to Rag2-/- immunodeficient mice that express Dsc3 and Dsg3), the presence of anti-Dsc3 autoantibodies is sufficient to induce the appearance of a pathological phenotype relatable to atypical pemphigus [17]. In addition, anti-Dsc3 antibodies in sera of pemphigus patients were proved to mainly recognize the extracellular 2 domain of Dsc3, and antibodies against extracellular 2 domain of Dsc3 have pathogenic roles on keratinocyte dissociation [18]. Although these studies proved the possible involvement of anti-Dsc3 antibodies in the pathogenesis of pemphigus, more deep studies are necessary to elucidate it clearly. ”